# Efficacy of Low-Dose Naltrexone and Predictors of Treatment Success or Discontinuation in Fibromyalgia and Other Chronic Pain Conditions: A Fourteen-Year, Enterprise-Wide Retrospective Analysis

**DOI:** 10.3390/biomedicines11041087

**Published:** 2023-04-03

**Authors:** C. Noelle Driver, Ryan S. D’Souza

**Affiliations:** Department of Anesthesiology and Perioperative Medicine, Division of Pain Medicine, Mayo Clinic, Rochester, MN 55905, USA

**Keywords:** naltrexone, chronic pain, fibromyalgia

## Abstract

Current pharmacologic treatments may provide limited analgesia in fibromyalgia and other chronic pain disorders. Low-dose naltrexone (LDN) has emerged as a potential analgesic option that has been minimally explored. This study aims to describe current real-world prescribing practices of LDN, to investigate if patients have a perceived benefit of LDN in treating pain symptoms and to identify predictors associated with a perceived benefit or discontinuation of LDN. We evaluated all outpatient prescriptions for LDN prescribed for any pain indication in the Mayo Clinic Enterprise from 1 January 2009 to 10 September 2022. A total of 115 patients were included in the final analysis. The patients were 86% female, had a mean age of 48 ± 16 years, and 61% of prescriptions were for fibromyalgia-related pain. The final daily dose of oral LDN ranged from 0.8 to 9.0 mg, while the most common dose was 4.5 mg once daily. Of patients who reported follow-up data, 65% reported benefit in their pain symptoms while taking LDN. Adverse effects were reported in 11 (11%) patients and 36% discontinued taking LDN by the most recent follow-up. Concomitant analgesic medications were used by 60% of patients and were not associated with perceived benefit nor discontinuation of LDN, including concomitant opioids. LDN is a relatively safe pharmacologic option that may benefit patients with chronic pain conditions and warrants further investigation in a prospective, controlled, and well-powered randomized clinical trial.

## 1. Introduction

Fibromyalgia is a prevalent musculoskeletal disorder affecting approximately 2–6.4% of people in the United States [1,2]. Further, chronic pain disorders affect approximately 100 million Americans, surpassing those who have heart disease or diabetes [3]. Symptoms of fibromyalgia include diffuse pain and stiffness in addition to fatigue, disrupted sleep, depression, and impaired cognition [1]. Patients with fibromyalgia and other chronic pain conditions have heightened sensitivity to both painful stimuli (hyperalgesia) and non-painful stimuli (allodynia).

Although the etiology of fibromyalgia is likely multifactorial (biological, environmental, and genetic factors), there is growing evidence for both central and peripheral biological immune-mediated mechanisms to explain these phenomena [4], which may inform new therapeutic targets. Activation of toll-like receptor 4 (TLR-4) in microglia and central nervous system neurons promotes proinflammatory cytokines which have been shown to be mediators of neuropathic pain [5]. These proinflammatory cytokines enhance excitatory tone in nociceptive neural networks, leading to heightened pain sensitivity in addition to fatigue, cognitive disruption, and mood and sleep disorders [6,7]. Predictors of fibromyalgia symptom severity such as tobacco use [8], hypovitaminosis D [9], exposure to various pharmaceuticals [10], and gender [11] have also been explored, highlighting the complex nature and pathophysiology of fibromyalgia.

Pharmacologic therapies for the treatment of fibromyalgia are limited in efficacy and are often associated with intolerable side effects. Analgesics that are currently approved by the Food and Drug Administration (FDA) for treatment of fibromyalgia include pregabalin, a gabapentinoid which acts at the alpha-2 voltage-dependent calcium channels, and duloxetine and milnacipran, two serotonin-norepinephrine reuptake inhibitors [12]. Conservative treatment options include physical therapy, exercise, acupuncture, and cognitive behavioral therapy. Interventional treatment options are often limited due to the diffuse manifestation of fibromyalgia. A common interventional option is trigger point injections with local anesthetic and/or steroid medication in areas that are painful in patients with fibromyalgia.

Naltrexone is another off-label pharmacologic option for chronic pain disorders that has been used sparsely in this population. Naltrexone is well-known as a non-selective opioid receptor antagonist used to treat opioid use disorder and alcohol use disorder. It has also been shown to have anti-inflammatory effects; it inhibits central nervous system proinflammatory cytokine activity via TLR-4 antagonism in microglia [13] and modulates mitochondrial apoptosis [14]. However, these processes are independent; at low concentrations, naltrexone has been shown to antagonize TLR-4 without antagonizing opioid receptors [13]. Additionally, naltrexone has anti-inflammatory effects at the periphery including suppression of tumor necrosis factor-alpha, interleukin-6, and many other inflammatory mediators [15].

Low-dose naltrexone (LDN) is commonly defined as oral dosages ranging from 1–5 mg daily but can go up to 4.5 mg twice per day [16,17,18,19]. This is a dose that is 5–10-fold lower than the typical dose of naltrexone used for opioid or alcohol use disorder. Naltrexone in these low doses is often made by specialized compounding pharmacies. Dosing frequency is often once-daily or twice-daily pills. Naltrexone may have variable and often opposing effects depending on the administered dose. Standard daily oral naltrexone doses of 50–100 mg result in opioid receptor antagonism and are used for alcohol and opioids use disorder [16]. Very low-dose naltrexone (0.001 mg to 1 mg) may have the same mechanism of action as LDN, and has been clinically utilized as add-on therapy to methadone detoxification taper [16]. Finally, ultra low-dose naltrexone (<0.001 mg) is involved in potentiation of opioid analgesia [16,20,21]. LDN has been trialed in patients with fibromyalgia as well as other chronic pain and autoimmune disorders including complex regional pain syndrome [22], Crohn’s disease [23,24], rheumatic diseases [25], multiple sclerosis [26], corneal neuralgia [27], burning-mouth syndrome [28], and painful diabetic neuropathy [29].

Despite growing evidence, data on the efficacy of LDN are limited to several small clinical trials and case reports. It is a generic, compounded medication with a lack of commercial interest among vendors in pursuing large clinical trials [18]. In this retrospective single cohort study, we describe patient characteristics, prescribing practices, and evidence of efficacy of LDN for adults with chronic pain conditions including fibromyalgia. We aim to investigate if LDN has perceived benefit in treating pain symptoms in a real-world setting and predictors associated with perceived benefit or discontinuation of LDN.

## 2. Materials and Methods

### 2.1. Population

This study was given exempt status by the Mayo Clinic Institutional Review Board (IRB Number 22-001214, granted 29 April 2022). The electronic medical records (EMR) from the Mayo Clinic Enterprise including three academic tertiary care centers (Rochester, Minnesota; Jacksonville, Florida; Scottsdale, Arizona) and all upper midwest community health systems were queried. All facilities use a common EMR (Epic Systems, Madison, WI, USA). All outpatient prescriptions containing the word “naltrexone” and oral dosages up to 10 mg daily from 1 January 2009 to 10 September 2022 were queried from the EMR and data on eligible patients were extracted. Patients were included if they were prescribed LDN for any pain indication (including fibromyalgia-related pain). Patients were excluded from further data extraction if they were prescribed LDN for non-pain indications such as obesity, autoimmune processes, and chronic fatigue. There were more prescriptions than corresponding patients due to duplications in the prescription data or refills.

### 2.2. Data Collection

The resulting patient charts were manually reviewed for data extraction. The following variables were collected: date of birth, gender, diagnosis associated with naltrexone prescription, first naltrexone order date, starting dose and frequency, titration interval, final dose and frequency, pain scores at time of and within 6 months of the prescription, reported adverse effects, discontinuation date, specialty of prescriber, concomitant analgesic medications, and patient report of the medication being beneficial or leading to moderate or substantial improvement in pain intensity. All records were queried for follow-up data up through the date of chart review in October 2022.

Concomitant analgesic medications were extracted according to the following definitions. Gabapentinoids included gabapentin and pregabalin. Serotonin-norepinephrine reuptake inhibitors (SNRI) included desvenlafaxine, duloxetine, milnacipran, and venlafaxine. Muscle relaxants included baclofen, cyclobenzaprine, and tizanidine. Opioids included hydrocodone, morphine, oxycodone, hydrocodone-acetaminophen, and oxycodone-acetaminophen combination products. No patients were prescribed codeine, fentanyl, methadone, or buprenorphine but these medications were also queried. Tramadol was considered separately from other opioids and SNRIs due to multiple mechanisms of action including µ-opioid agonism, serotonin reuptake antagonism, and norepinephrine reuptake antagonism, as well as other targets. Tricyclic antidepressants included amitriptyline and nortriptyline. Data on acetaminophen, non-steroidal antiinflammatory medication, and topical medication use were not recorded. Additionally, data on historical or concomitant interventional pain procedures or injections such as botulinum toxin, local anesthetics or steroids were not recorded.

### 2.3. Statistical Analyses

For descriptive characteristics, continuous variables were summarized by the mean, standard deviation, and range. Categorical variables were summarized as frequency count and percentages.

Univariate analyses were conducted to assess factors associated with (1) LDN being perceived as beneficial for pain-related symptoms, and (2) discontinuation of LDN among patients for whom these data were available. Binomial regression models were performed to determine the odds ratios, 95% confidence intervals and *p*-values for gender, age, concomitant medications, and final LDN dose on these two outcomes. *p*-values were determined by Wald tests and values < 0.05 were considered to be significant. Due to small sample size, the primary focus was on univariate analyses. Any variables with significant *p*-values in the univariate models were included in multivariate binomial regression models for the same outcome variables. All analyses were performed in R v.4.0.5.

### 2.4. Subgroup Analysis

Additional analysis was performed on patients subgrouped into those prescribed LDN who were not on any concomitant analgesic medications versus those prescribed LDN plus concomitant analgesic medications. For descriptive characteristics, continuous variables were summarized by the mean, standard deviation, and range. Categorical variables were summarized as frequency count and percentages.

Descriptive statistics were compared between these two groups by t-tests for continuous variables and Χ^2^ tests for categorical variables. *p*-values < 0.05 were considered to be significant differences between groups.

## 3. Results

### 3.1. Data Collection

Among 32.5 million total outpatient prescriptions in the EMR database (accessed on 20 August 2022), 203 prescriptions were for naltrexone at doses less than or equal to 10 mg daily between 1 January 2009 and 15 October 2022. This corresponded to a total of 104 patients. These 104 patients were combined with an additional 30 patients previously extracted by the senior author (RSD) with prescriptions for LDN. Complete chart review for the variables mentioned above was performed on these 134 patients. After 19 patients were removed due to naltrexone prescriptions for non-pain indications (including obesity, autoimmune processes, chronic fatigue, dysautonomia, and alcohol cessation), there was a final total sample of 115 patients (Figure 1).

Pain scores at the time and within six months of the initial LDN prescription were queried and recorded. However, these data were sparse and if they were recorded, they were associated with other encounters such as a visit to an emergency department or pre-/post-procedures. They did not represent average daily pain scores and thus were not included in further analysis.

### 3.2. Patient and Prescription Characteristics

Table 1 describes summary data for patient and prescription characteristics. Average age at the time of the first LDN prescription was 48 ± 16 years and patients were predominantly female (86%). Prescribers were from the following medical specialties: integrative medicine or fibromyalgia-specific clinics (29%), primary care (19%), neurology (19%), or pain medicine (13%). Indications for LDN included fibromyalgia (61%), chronic pain (11%), myalgic encephalomyelitis/chronic fatiguing syndrome (8%), and other chronic pain conditions.

Table 2 describes LDN usage. A majority (94%) of patients started LDN after receiving their prescription. Of 68 patients with follow-up information, 44 (65%) reported a perceived benefit in terms of their pain symptoms in addition to other symptoms (e.g., fatigue, brain fog, sleep). Average daily starting dose was 2.9 mg but ranged from 0.3 to 6.0 mg with a final average daily dose of 4.2 mg (range 0.8 to 9.0 mg). The most common prescription regimen was a starting dose of 1.5 mg daily, uptitrated to 4.5 mg daily in increments of 1.5 mg over the course of 2–4 weeks as tolerated. A majority of patients (67 patients, 60%) were on at least one concomitant analgesic, including 37% on pregabalin or gabapentin, 16% on a SNRI, and 11% on chronic opioid medications. Thirty-six patients (37%) discontinued LDN and among those who discontinued it, the average trial period was 179 days and ranged from 7 to 720 days.

### 3.3. Adverse Effects

Only 11 patients reported adverse effects of LDN (Table 3) which precipitated discontinuation of the medication. Most common adverse effects were nausea/vomiting, headaches, and a perceived sense of increased anxiety. Some patients reported more than one adverse effect.

### 3.4. Logistic Regression

Univariate logistic regression models were used assess whether age, gender, concomitant analgesic medications, or final LDN dose were predictive of (1) LDN being reported as beneficial and (2) LDN discontinuation (Table 4). Each line of Table 4 represents an individual univariate model assessing the variable’s odds on the outcomes of perceived benefit or discontinuation, respectively. The total *n* for each model represents the number of patients with complete data for each pair of variables. None of these models showed significant associations and thus no multivariate models were performed. Due to small sample size, these results are likely underpowered if true associations exist.

### 3.5. LDN-Only versus LDN-Plus-Concomitant Analgesics Subgroup Analysis

A subgroup analysis was performed on the 45 patients prescribed LDN who were not on any concomitant analgesic medications versus the 67 patients prescribed LDN with concomitant analgesics (Table 5). There were no significant differences found between these two groups in terms of age, gender, indication for LDN, started LDN prescription, starting and final doses of LDN, perceived benefit, reported side effects, discontinuation of LDN, and length of trial period via unpaired t-tests or Χ^2^ tests. Although not statistically significant, it is interesting to note a larger percentage of patients on concomitant analgesics reported side effects compared to the LDN-only group (14% vs. 8%, *p* = 0.5). This may reflect pharmacologic interactions.

## 4. Discussion

This study identified that a majority of patients (65%) who took LDN for any pain indication reported a benefit in their pain symptoms. A significant portion of our cohort comprised patients with fibromyalgia, which is an often challenging condition to diagnose and treat. Potential mechanisms that may explain the analgesic benefit from naltrexone at low doses ranging from 1–5 mg include glial modulation and antagonism of TLR-4 that is implicated in generating inflammatory end-products [16]. Subsequently, LDN may lead to an attenuated pro-inflammatory profile after administration. Our clinical findings of benefit are concordant with other published studies including a single-blind crossover trial of 10 patients with fibromyalgia which reported that LDN reduced symptoms of fibromyalgia by at least 30% compared to placebo, and that mechanical and heat pain thresholds were improved [30]. Similarly, a double-blind, crossover, counterbalanced study of 30 women with fibromyalgia found 57% of women to have significant reduction in pain with LDN treatment [31]. Another matched retrospective cohort study was conducted with 36 patients receiving 4.5 mg LDN daily and 42 control patients matched by indication (including both inflammatory and neuropathic conditions), age and gender. This study compared pain scores at baseline to pain scores at the final documented visit and reported approximately 42% pain reduction with LDN among patients with neuropathic-related pain compared to control patients [32]. While large multicenter clinical trials are warranted to better quantify estimates of efficacy, these data support an overall favorable efficacy profile for LDN for chronic pain disorders.

The current study also analyzed for predictors of treatment benefit and predictors of treatment discontinuation, including gender, age, and administration of certain analgesics (gabapentinoids, tricyclic antidepressants, tramadol, SNRIs, muscle relaxants, and opioids), and LDN dose. Although the analysis did not identify statistical significance for any predictor variable, future studies are warranted to validate these data. Future studies may also explore other predictor variables such as smoking status, alcohol use, history of substance abuse, and other potential negative predictors.

In our current study, only 11 (11%) patients reported side effects from LDN, none of which were severe or life-threatening. Previous research has suggested side effects of LDN treatment are uncommon and mild, including vivid dreams, headaches, anxiety, and tachycardia [17]. No serious side effects have been reported and there does not appear to be a risk of withdrawal, dependence, or tolerance [17]. Overall, LDN appears to be safe [33,34,35].

It should be noted that 11% of patients in our cohort were on concomitant chronic opioid medications. Many clinicians are hesitant to deliver LDN in those with opioids due to concern that LDN is a µ-opioid receptor antagonist. Our regression analysis suggests that concomitant opioids and LDN is not predictive of discontinuation of LDN therapy (*p* = 0.38). Evidence from animal models suggests that LDN administered with oxycodone attenuates neuropathic pain and reduces tolerance to the oxycodone alone [21]. Furthermore, two clinical trials evaluated the combination of oxycodone and LDN on chronic back pain and osteoarthritis. The groups receiving naloxone had 12% lower daily opioid consumption compared to those receiving opioids alone and further reported better pain relief [20,36]. At low doses of naltrexone, we do not see opioid withdrawal symptoms and interestingly, patients with co-administration of LDN with oxycodone had lower rates of withdrawal symptoms after discontinuation [18,36]. Thus, current chronic opioid use should not contraindicate a trial of LDN. Importantly, physicians should consider offering LDN as part of a multimodal analgesic regimen that includes medications with differing mechanisms of action [37,38,39].

Most prescribing physicians were from integrative medicine/fibromyalgia clinics (29%), primary care (19%), and neurology (19%). Pain medicine physicians comprised only a small portion of prescribing physicians (13%). There may be several potential reasons for this finding. Firstly, pain medicine physicians are trained in performing a variety of interventional procedures. These include ultrasound-guided injections, fluoroscopic-guided injections, and neuromodulation interventions such as spinal cord stimulation and peripheral nerve stimulation. Therefore, pain medicine physicians may be inclined to offer these interventional modalities before trialing analgesic medications with limited evidence. Secondly, many specialties may be unfamiliar with LDN as an analgesic due to its limited evidence base in chronic pain treatment. Finally, there may also be concerns for its side effect profile and its potential interaction with opioids medications.

The authors query possible explanations for the high rate of discontinuation (36%) of LDN in the current study. In most cases, a specific reason for discontinuation was not provided or clarified in the patient’s EMR. Although most of these patients likely did not experience meaningful analgesia from LDN, it is plausible that a portion may have obtained pain relief and may have discontinued the medication after achieving analgesia and attenuation of central sensitization.

This study is limited by the constraints of retrospective EMR data. Average daily pain scores at baseline and within 6 months trial of LDN would have provided valuable efficacy data. Because pain scores were unavailable or unreliable, we chose to focus on evidence via follow-up notes or patient communication with explicit mention of benefit. However, many patients had no follow-up data and thus our estimates may include selection or reporting bias. Long-term effects of LDN could not be ascertained from this limited review and duration of concomitant medications likewise could not be extracted from the EMR as it was unclear when patients may have discontinued them.

We additionally were underpowered to assess associations of patient factors with perceived LDN benefit or discontinuation in the binomial models. Since there was no comparative arm present (e.g., placebo or another medication), conclusions from the study are limited and the authors can not exclude the possibility of a placebo effect [40]. Our sample has heterogeneous indications; targeted studies by indication may further elucidate which patients may be more likely to benefit from LDN use. Future studies should also nuance the use of this medication in different patients (e.g., weight-dependence or concomitant medications) as well as study interactions with other medications.

With the favorable LDN efficacy and safety data we have thus far, and in the setting of limited non-addictive pharmacologic options for treatment of chronic pain, LDN can be considered for a trial in patients with fibromyalgia and other chronic pain conditions. Future randomized, placebo-controlled trials are warranted to better assess patient-related factors associated with efficacy. The current study was focused on treatment success or change in pain intensity. Other patient-reported outcomes such as physical function, disability, pain interference, quality of life, mental health outcomes, and satisfaction rates should also be explored in formal surveys. Studies should also explore the mechanisms of action of LDN that may explain its therapeutic attenuation of central sensitization and chronic pain. Additional analgesics such as methadone [41,42] and ketamine [12] have also been noted to attenuate central sensitization and the wind-up phenomenon [43,44]. Studies should explore if there is an additive or synergistic effect on attenuation of central sensitization when these analgesics are concomitantly administered with LDN therapy. Finally, our study identified that there was a wide range of final LDN oral doses administered to patients, ranging from 0.9 mg daily to 9 mg daily (in single or divided doses). The most common regimen was 4.5 mg once daily, which is concordant with prior trials. However, a dose-finding study and dose-response study are lacking and future investigation is warranted to identify the optimal dosing of LDN.

## 5. Conclusions

LDN may provide meaningful pain relief in most patients with fibromyalgia and other refractory chronic pain syndromes. This study did not identify predictors of treatment success or discontinuation. Generally, adverse effects from LDN are mild and infrequently reported. Other analgesic medications, including opioids, may be co-administered concomitantly with LDN. Further investigation in a prospective, controlled, and well-powered randomized clinical trial is warranted.

## Figures and Tables

**Figure 1 biomedicines-11-01087-f001:**
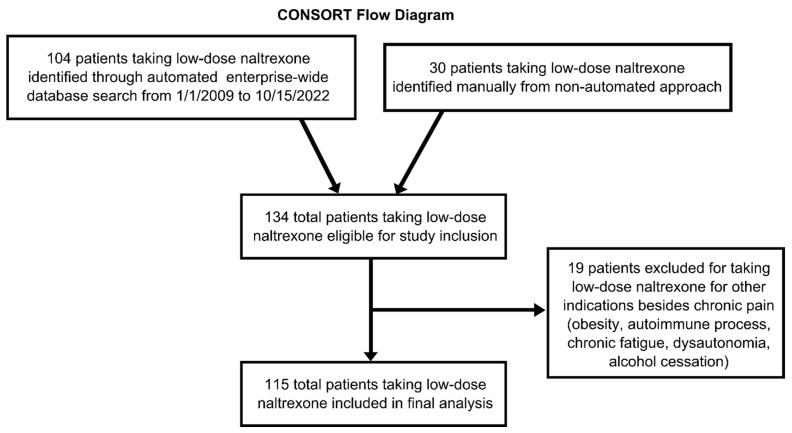
CONSORT Diagram: patient inclusion and exclusion process are displayed.

**Table 1 biomedicines-11-01087-t001:** Patient and Prescription Characteristics (*n* = 115).

	Sample with Data (*n*)	Mean (SD) or *n* (%)	
Age (y)	98	48	16
Gender	115		
Female		99	86%
Male		14	12%
Non-binary		1	1%
Transgender		1	1%
Specialty of Prescriber	115		
Integrative Medicine/Fibromyalgia Clinic		33	29%
Primary Care/Family Medicine		22	19%
Neurology		22	19%
Pain Medicine		15	13%
Gastroenterology		3	3%
Rheumatology		3	3%
PM&R		2	2%
Unknown		15	13%
Indication	115		
Fibromyalgia		70	61%
Chronic Pain		13	11%
ME/CFS		9	8%
Multiple Sclerosis		7	6%
Myofascial Pain Syndrome		4	3%
Chronic Abdominal Pain		3	3%
CRPS		2	2%
Chronic Neck Pain		2	2%
Chronic Low Back Pain		1	1%
Chronic Leg Pain		1	1%
Postprandial Pain		1	1%
Chronic Migraine		1	1%
Pityriasis Rubra Pilaris Type 2 Adult		1	1%

Abbreviations: y = years, PM&R = Physical Medicine and Rehabilitation, ME/CFS = myalgic encephalomyelitis/chronic fatigue syndrome, CRPS = complex regional pain syndrome, SD = standard deviation.

**Table 2 biomedicines-11-01087-t002:** Low-Dose Naltrexone Usage (*n* = 115).

	Sample with Data (*n*)	Mean (SD) or *n* (%)	
Started Prescription	98	92	94%
Starting Naltrexone Daily Dose (mg)	107	2.9	1.6
(Minimum–Maximum)		(0.3–6.0)	
Final Naltrexone Daily Dose (mg)	109	4.2	1.0
(Minimum–Maximum)		(0.8–9.0)	
Perceived Benefit	68	44	65%
Reported Side Effects	98	11	11%
Concomitant Medications	112		
Any Analgesic		67	60%
Gabapentinoid		41	37%
SNRI		18	16%
Muscle Relaxant		17	15%
Opioid		12	11%
Tramadol		12	11%
Tricyclic Antidepressant		9	8%
Discontinued Naltrexone	97	36	37%
Length of Trial (days)	36	179	160
(Minimum–Maximum)		(7–720)	

Abbreviations: mg = milligrams, SD = standard deviation, SNRI = selective serotonin and norepinephrine reuptake inhibitor.

**Table 3 biomedicines-11-01087-t003:** Adverse Effects (*n* = 11 patients).

	*n*
Nausea/vomiting	4
Headaches	3
Anxiety	3
Restlessness	2
Dizziness	2
Insomnia	2
Sweats/hot flashes	2
Fatigue	1
Chills	1
Nightmares	1
Decreased appetite	1
Leg pain	1
Bloating	1
Constipation	1
Depression	1
Hypertension	1

**Table 4 biomedicines-11-01087-t004:** Univariate Binomial Models.

Odds of Perceived Benefit				
Variable	*n*	Odds Ratio	(95% CI)	*p*-Value
Gender	68	1.06	(0.25–3.95)	0.94
Age	59	1.00	(0.97–1.04)	0.82
Any Concomitant Analgesic	68	0.87	(0.30–2.39)	0.78
Gabapentinoid	68	1.14	(0.41–3.36)	0.80
Opioid	68	0.70	(0.14–3.83)	0.66
Tricyclic Antidepressant	68	0.24	(0.03–1.32)	0.11
Tramadol	68	2.08	(0.45–14.81)	0.39
SNRI	68	0.72	(0.20–2.71)	0.61
Muscle Relaxant	68	0.90	(0.20–4.72)	0.89
Final LDN Dose	67	1.16	(0.70–2.00)	0.56
**Odds of Discontinuation**				
**Variable**	* **n** *	**Odds Ratio**	**(95% CI)**	***p*-value**
Gender	97	1.22	(0.39–4.21)	0.74
Age	82	1.01	(0.98–1.04)	0.51
Any Concomitant Analgesic	97	1.59	(0.68–3.82)	0.29
Gabapentinoid	97	1.13	(0.48–2.64)	0.78
Opioid	97	1.81	(0.47–6.97)	0.38
Tricyclic Antidepressant	97	1.78	(0.40–8.00)	0.44
Tramadol	97	1.81	(0.47–6.97)	0.38
SNRI	97	1.89	(0.63–5.68)	0.25
Muscle Relaxant	97	1.16	(0.36–3.53)	0.80
Final LDN Dose	96	0.77	(0.48–1.19)	0.26

Abbreviation: CI = confidence interval, LDN = low-dose naltrexone, SNRI = serotonin-norepinephrine reuptake inhibitor.

**Table 5 biomedicines-11-01087-t005:** LDN-Only vs. LDN-plus-Concomitant Analgesic Subgroup Characteristics.

	LDN-Only Subgroup (*n* = 45)			LDN-Plus-Concomitant Analgesics Subgroup (*n* = 67)				
	Sample with Data (*n*)	Mean (SD) or *n* (%)		Sample with Data (*n*)	Mean (SD) or *n* (%)		T-Test or Χ^2^	*p*-Value
Age (y)	41	45	14	56	50	15	−1.60	0.11
Gender	45			67			3.98	0.05
Female		43	96%		53	79%		
Male		1	2%		13	19%		
Non-binary		1	2%		0	0%		
Transgender		0	0%		1	1%		
Indication	45			67			9.66	0.72
Fibromyalgia		30	67%		38	57%		
Chronic Pain		4	9%		8	12%		
ME/CFS		3	7%		6	9%		
Multiple Sclerosis		3	7%		3	4%		
Myofascial Pain Syndrome		0	0%		4	6%		
Chronic Abdominal Pain		1	2%		2	3%		
CRPS		1	2%		1	1%		
Chronic Neck Pain		1	2%		1	1%		
Chronic Low Back Pain		0	0%		1	1%		
Chronic Leg Pain		0	0%		1	1%		
Postprandial Pain		1	2%		0	0%		
Chronic Migraine		1	2%		0	0%		
Pityriasis Rubra Pilaris Type 2		0	0%		1	1%		
Started LDN Prescription	40	38	95%	56	54	96%	<0.01	>0.99
Starting LDN Daily Dose (mg)	43	3.2	1.6	63	2.7	1.6	1.46	0.15
(Minimum–Maximum)		(0.5–6.0)			(0.3–4.5)			
Final LDN Daily Dose (mg)	44	4.3	1.1	64	4.2	0.9	0.76	0.45
(Minimum–Maximum)		(1.0–9.0)			(0.8–6.0)			
Perceived Benefit	27	18	67%	41	26	63%	<0.01	>0.99
Reported Side Effects	40	3	8%	56	8	14%	0.46	0.50
Discontinued LDN	39	12	31%	58	24	41%	0.72	0.40
Length of Trial (days)	12	171	151	20	182	167	−0.17	0.87
(Minimum–Maximum)		(10–360)			(7–720)			

Abbreviations: LDN = low-dose naltrexone, mg = milligrams, SD = standard deviation, y = years, ME/CFS = myalgic encephalomyelitis/chronic fatigue syndrome, CRPS = complex regional pain syndrome.

## Data Availability

The data presented in this study are available on request from the corresponding author.

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
