# Peer review of "Efficacy of Low-Dose Naltrexone and Predictors of Treatment Success or Discontinuation in Fibromyalgia and Other Chronic Pain Conditions: A Fourteen-Year, Enterprise-Wide Retrospective Analysis"

_biomedicines, 2023, doi:10.3390/biomedicines11041087_

Round 1

Reviewer 1 Report

The abstract lacks information about the route of administration and daily (?) dose(s?) of LDN.

Throughout the work, the authors do not indicate whether the dose is per patient or per kg, whether a single or daily dose. The route of administration is not specified as well.

The standard dose of NAL is 25-50-100 mg. How were patients given low doses of LDN?

Did the authors consider depot treatment with NAL (i.m. injections)? Such treatment could be much more effective, and the drug administration could be reduced to one monthly dose.

The authors do not specify the time of treatment of patients, whether the drug was effective immediately after the begining, after what time the surveys were carried out, and the long-term effects of treatment.

Analyzing the data presented at the beginning of the "Results", it appears that 100 patients received 200 prescriptions, which means that one patient got, on average, two prescriptions. So what was the treatment time? Why did the treatment not include a long-term approach?

In the chapter "2.1. Population," the authors focus on the NAL selection criterion. Still, they do not specify how the process of confirming and including/excluding patients was carried out due to the type of pain (fibromyalgia and others).

Table 2 shows that out of 115 patients, 112 received other medications. I understand that it was impossible to obtain patients treated with LDN only. Still, such a group would seem logical to evaluate, and groups receiving treatment without LDN, which seems to be common practice as a reference group. Such a comparison group certainly exists in the database of 32 million prescriptions. There is also no placebo group to draw complete conclusions.

Why, in table 2, did the authors not provide the number of patients in the groups receiving other drugs? It is given N=112, but there are data for individual drugs.

Why do the authors divide Opioids and Tramadol separately when both drugs belong to the same class?

Why did the authors not specify what opioid drugs the patients were taking? The effects of Fentanyl, codeine, and buprenorphine are entirely different – functional and on the molecular level as well. Perhaps such information would be significant. The duration of treatment is also important because, at the receptor level, we observe different (opposite) changes after one administration, weekly and several years.

What did the authors mean by muscle relaxants? The group is very diverse, botox, clonazepam, baclofen, anti-Parkinson drugs... Their mechanism of action is very different and should not be treated together.

In table 3, N is the number of patients. Meanwhile, in table 4, there is also N, which suggests that it also refers to the number of patients. How should these values be interpreted? All patients were in some age and gender, so shouldn't N be 115 in those lines? Why is N=68 for all drugs when in table 2 N=112 for drugs? Were all patients taking all these drugs together? Please explain.

Citations need to be corrected in style consistent with the rules of the journal [1], [1,2], [1,2,3-5].

Author Response

1) The abstract lacks information about the route of administration and daily (?) dose(s?) of LDN.

Author Response: Firstly, we would like to forward the reviewer our utmost appreciation and time for the feedback. We agree with this comment. We have amended the abstract to read, “The final daily dose of oral LDN ranged from 0.8 to 9.0 mg, while the most common dose was 4.5 mg once daily.”

2) Throughout the work, the authors do not indicate whether the dose is per patient or per kg, whether a single or daily dose. The route of administration is not specified as well.

Author Response: Thank you for this feedback and for the opportunity to clarify. All doses of LDN were prescribed per patient at once daily dosing or up to twice daily dosing. There were no weight-based prescriptions. We also clarified throughout the manuscript that LDN is prescribed as oral administration only.

Edit to the manuscript (Line 67): “Low-dose naltrexone (LDN) is commonly defined as oral dosages ranging from 1-5 mg daily but can go up to 4.5 mg twice per day [16-19]. This is a dose that is 5-10-fold lower than the typical dose of naltrexone used for opioid or alcohol use disorder. Naltrexone in these low doses is often made by specialized compounding pharmacies. Dosing frequency is often one daily or twice-daily pills.”

3) The standard dose of NAL is 25-50-100 mg. How were patients given low doses of LDN?

Author Response: Thank you for this question and for the opportunity to clarify. Naltrexone at these low doses is only currently available through specialized compounding pharmacies, where pills at doses such as 1.5 mg each are made. We have added the following sentence to the introduction to clarify this for readers (Like 67): “Naltrexone in these low doses is often made by specialized compounding pharmacies. Dosing frequency is often one daily or twice-daily pills.”

Further, we also clarified that higher doses of naltrexone as indicated by the reviewer are offered for other clinical indications (e.g. alcohol and opioid use disorder) and not for analgesia (Line 72): “Standard daily oral naltrexone doses of 50-100 mg result in opioid receptor antagonism and are used for alcohol and opioids use disorder [16].”

4) Did the authors consider depot treatment with NAL (i.m. injections)? Such treatment could be much more effective, and the drug administration could be reduced to one monthly dose.

Author Response: Thank you for this feedback. In our retrospective review, we did not find LDN prescribed as a depot injection. To our knowledge, all clinical trials of LDN thus far have used daily oral administrations. Monthly injections could be considered in future prospective research. Further, depot formulations of standard naltrexone (not LDN) are indicated for opioid use disorder and alcohol use disorder.

5) The authors do not specify the time of treatment of patients, whether the drug was effective immediately after the beginning, after what time the surveys were carried out, and the long-term effects of treatment.

Author Response: Thank you for this important feedback.  As a retrospective review of patients with heterogeneous pain indications and prescriptions from prescribers across multiple states and specialties, our data was limited to what were available in the patient charts, either from follow-up patient messages or follow-up appointments. There were no formal effectiveness surveys conducted. Due to these limited data, we focused analysis on a simplified, “perceived benefit” variable which was recorded at any point during the period when follow-up data was available (up to present date of chart review).  The following sentence has been added to the Methods section to clarify this (Line 110): “All records were queried for follow-up data up through the date of chart review in October 2022.” There were only 36 patients with data available in the records that indicated discontinuation, and the trial period lengths were recorded for these patients (average 179 days, range 7 – 720 days). We have added the following sentence to the discussion as well (Line 299): “Long-term effects could not be ascertained from this limited review.”

6) Analyzing the data presented at the beginning of the “Results”, it appears that 100 patients received 200 prescriptions, which means that one patient got, on average, two prescriptions. So what was the treatment time? Why did the treatment not include a long-term approach?

Author Response: Thank you for this feedback and for the opportunity to clarify. Prescriptions were initially queried as a more sensitive approach to capture as many potential patients on LDN as possible, which were subsequently manually reviewed. Patients may have had more than one prescription either due to duplications in the data or refills. The prescription data do not adequately capture treatment time. The following sentence has been added to the Methods section to clarify this for readers (Line 101): “There were more prescriptions than corresponding patients due to duplications in the prescription data or refills.”

7) In the chapter “2.1. Population,” the authors focus on the NAL selection criterion. Still, they do not specify how the process of confirming and including/excluding patients was carried out due to the type of pain (fibromyalgia and others).

Author Response: Thank you for this feedback. We agree the process of inclusion/exclusion of patients warrants further clarification. We have added the following sentences to section 2.1: “Patients were included if they were prescribed LDN for any pain indication (including fibromyalgia-related pain). Patients were excluded from further data extraction if they were prescribed LDN for non-pain indications such as obesity, autoimmune processes, and chronic fatigue.”

8) Table 2 shows that out of 115 patients, 112 received other medications. I understand that it was impossible to obtain patients treated with LDN only. Still, such a group would seem logical to evaluate, and groups receiving treatment without LDN, which seems to be common practice as a reference group. Such a comparison group certainly exists in the database of 32 million prescriptions. There is also no placebo group to draw complete conclusions.

Author Response:  Thank you for this comment and for the opportunity to explore this question further. We agree with this comment and added an additional subgroup analysis comparing the descriptive statistics between LDN-only patients (n=45) vs LDN-plus-concomitant analgesics (n = 67) groups. Please see section 3.4 of the results and a new Table 5.  We agree placebo-controlled, and non-LDN treated pain would serve as good reference groups in prospective analysis. Given the variety of pharmacologic, behavioral and interventional treatments available for pain indications and lack of formal follow-up period or survey, we do not see how a non-LDN reference group could be identified with the current study and do recommend future clinical trials to address these limitations.

9) Why, in table 2, did the authors not provide the number of patients in the groups receiving other drugs? It is given N=112, but there are data for individual drugs.

Author Response: Thank you for this question and for the opportunity to clarify.  We agree it is helpful to clarify the total number of patients with any concomitant analgesic (67 patients, 60%). We have added this to Table 2 and clarified in the text.

10) Why do the authors divide Opioids and Tramadol separately when both drugs belong to the same class?

Author Response: Thank you for this feedback. We added a paragraph explaining which medications were included in each class. We separated opioid agonists (hydrocodone, oxycodone, etc) from tramadol due to its effects on multiple receptors. This is now clarified in the following sentences (Line 115): “Opioids included hydrocodone, morphine, oxycodone, hydrocodone-acetaminophen and oxycodone-acetaminophen combination products…. Tramadol was considered separately from other opioids and SNRIs due to multiple mechanisms of action including µ-opioid agonism, serotonin reuptake antagonism and norepinephrine reuptake antagonism, as well as other targets.”

11) Why did the authors not specify what opioid drugs the patients were taking? The effects of Fentanyl, codeine, and buprenorphine are entirely different – functional and on the molecular level as well. Perhaps such information would be significant. The duration of treatment is also important because, at the receptor level, we observe different (opposite) changes after one administration, weekly and several years.

Author Response: Thank you for this feedback. We added a paragraph explaining which medications were included in each class (Line 115): “Opioids included hydrocodone, morphine, oxycodone, hydrocodone-acetaminophen and oxycodone-acetaminophen combination products. No patients were prescribed codeine, fentanyl, methadone, or buprenorphine but these medications were also queried. Tramadol was considered separately from other opioids and SNRIs due to multiple mechanisms of action including µ-opioid agonism, serotonin reuptake antagonism and norepinephrine reuptake antagonism, as well as other targets.”  We additionally added as a limitation that duration of concomitant medications could not be extracted as it was unclear when patients may have discontinued them based on retrospective chart review.

12) What did the authors mean by muscle relaxants? The group is very diverse, otox, clonazepam, baclofen, anti-Parkinson drugs… Their mechanism of action is very different and should not be treated together.

Author Response: Thank you for this feedback! We have clarified which medications were included in the muscle relaxants group with the following sentence (Line 115): “Muscle relaxants included baclofen, cyclobenzaprine and tizanidine.”

13) In table 3, N is the number of patients. Meanwhile, in table 4, there is also N, which suggests that it also refers to the number of patients. How should these values be interpreted? All patients were in some age and gender, so shouldn’t N be 115 in those lines? Why is N=68 for all drugs when in table 2 N=112 for drugs? Were all patients taking all these drugs together? Please explain.

Author Response: Thank you for these questions and for the opportunity to clarify. In table 3, N = 11 as this is the total number of patients who reported adverse effects. In table 4, the Ns refer to the number of patients in each univariate model (each line). N = 68 for gender because this is the intersection between patients for whom we have data for gender (115) and data for perceived benefit (68). Likewise, N = 59 for age because this is the intersection between patients for whom we have data for age (98) and perceived benefit (68). Similarly for each of the drugs, we added the following sentences to further clarify this in the text for readers (Line 202): “Each line of Table 4 represents an individual univariate model assessing the variable’s odds on the outcomes of perceived benefit or discontinuation, respectively. The total N for each model represents the number of patients with complete data for each pair of variables.”

14) Citations need to be corrected in style consistent with the rules of the journal [1], [1,2], [1,2,3-5].

Author Response: Thank you. We updated the formatting of the references to comply with the rules of the journal.

Reviewer 2 Report

This paper approaches a relevant issue. However, the English needs a  review. In addition, there is a need to improve fluency of the text, especially regarding the connection between adjacent ideas. Overall, readability is low and requires attention from the authors

·      Please state a clear objective for the work in the abstract as well as in the main text. Try to improve the clarity of the introduction. 

Author Response

1) This paper approaches a relevant issue. However, the English needs a review. In addition, there is a need to improve fluency of the text, especially regarding the connection between adjacent ideas. Overall, readability is low and requires attention from the authors. Please state a clear objective for the work in the abstract as well as in the main text. Try to improve the clarity of the introduction. 

Author Response: Firstly, we would like to forward the reviewer our utmost appreciation and time for the feedback. We have added the following sentence to the abstract to clarify the objective: “This study aims to describe current real-world prescribing practices of LDN, to investigate if LDN has perceived benefit in treating pain symptoms and to identify predictors associated with perceived benefit or discontinuation of LDN.”  We have additionally reviewed the introduction and entire manuscript for fluidity and clarify and made changes throughout.

Round 2

Reviewer 1 Report

The authors have introduced the suggested corrections. The work has been improved and is suitable for publication in its present form.